

# Windows malware detection based on static analysis with multiple features

Muhammad Irfan Yousuf[1], Izza Anwer[2], Ayesha Riasat[3], Khawaja Tahir Zia[1] and Suhyun Kim[4]

[1] Department of Computer Science, University of Engineering and Technology Lahore, Lahore, Pakistan
[2] Department of Transportation Engineering and Management, University of Engineering and Technology Lahore, Lahore, Pakistan
[3] Department of Basic Sciences and Humanities, University of Engineering and Technology Lahore, Lahore, Pakistan
[4] Centre for Artificial Intelligence, Korea Institute of Science and Technology, Seoul, Republic of Korea

## ABSTRACT

Malware or malicious software is an intrusive software that infects or performs harmful activities on a computer under attack. Malware has been a threat to individuals and organizations since the dawn of computers and the research community has been struggling to develop efficient methods to detect malware. In this work, we present a static malware detection system to detect Portable Executable (PE) malware in Windows environment and classify them as benign or malware with high accuracy. First, we collect a total of 27,920 Windows PE malware samples divided into six categories and create a new dataset by extracting four types of information including the list of imported DLLs and API functions called by these samples, values of 52 attributes from PE Header and 100 attributes of PE Section. We also amalgamate this information to create two integrated feature sets. Second, we apply seven machine learning models; gradient boosting, decision tree, random forest, support vector machine, K-nearest neighbor, naive Bayes, and nearest centroid, and three ensemble learning techniques including Majority Voting, Stack Generalization, and AdaBoost to classify the malware. Third, to further improve the performance of our malware detection system, we also deploy two dimensionality reduction techniques: Information Gain and Principal Component Analysis. We perform a number of experiments to test the performance and robustness of our system on both raw and selected features and show its supremacy over previous studies. By combining machine learning, ensemble learning and dimensionality reduction techniques, we construct a static malware detection system which achieves a detection rate of 99.5% and error rate of only 0.47%.

# INTRODUCTION

Malicious software, commonly called malware, can be classified into viruses, worms, Trojans, spyware, ransomware, logic bomb, etc. based on their behavior and characteristics (*Gibert, Mateu & Planes, 2020*). Computer malware pose a major threat to computer and network security. This is the reason that research on developing new systems to detect malware is a hot topic in data mining, machine learning, and deep

Corresponding author
Muhammad Irfan Yousuf,
irfan.yousuf@uet.edu.pk,
dr.yousuf.irfan@gmail.com

learning. Our work is aimed at developing a static malware detection system to detect Portable Executable (PE) malware using multiple features. We not only extract multiple features from PE malware but also combine these features to create integrated features in a bid to improve the accuracy of our malware detection system. Presumably, the multiple and integrated features used in this work have never been considered together in detecting malware.

Most commercial anti-virus software rely on signature-based detection of malware, however, it is not effective against unknown malware or zero-day attacks. In the last decade or so, the research on malware detection has focused on finding generalized and scalable features to identify previously unknown malware and counter zero-day attacks effectively (*Guo, 2023*). There are two basic types of malware analyses; static analysis and dynamic analysis (*Damaševičius et al., 2021*). In static analysis, features are extracted from the code and structure of a program without actually running it whereas in dynamic analysis features are gathered after running the program in a virtual environment.

During the last decade, machine learning has solved many problems in different sectors including cyber security. It is now believed that AI-powered anti-virus tools can help in detecting zero-day attacks (*Alhaidari et al., 2022*). A typical machine learning workflow in detecting malware involves data collection, data cleaning and pre-processing, building and training models, validating, and deploying into production. In this regard, the success of supervised machine learning models depends on two factors: (1) the amount of labeled data used to train the model and (2) the features extracted from the malware. There have been numerous studies (*Sharma, Rama Krishna & Sahay, 2019*; *Chowdhury, Rahman & Islam, 2017*; *Kim et al., 2021*; *Patidar & Khandelwal, 2019*; *Zhang, Kuo & Yang, 2019*) on static malware analysis using machine learning but most of these studies train their models on one or two types of features and have their own limitations.

In this work, we propose a malware detection system for detecting Portable Executable (PE) malware based on static analysis with multiple features. We extract four types of feature sets and also merge them to create two additional feature sets. The research contributions made by the authors are listed below.

1. Collection of latest samples to create a new dataset of PE malware and benign files.
2. We extract four feature sets including the list of imported DLLs and API functions called by these samples, values of 52 attributes from the PE Header and 100 attributes of the PE Section.
3. We merge extracted features for creating new integrated features of PE samples.
4. Comprehensive analysis and evaluation of different machine learning classifiers, ensemble learning and feature selection techniques to maximize the malware detection rate.

The remainder of this article is organized as follows. We discuss the some previous studies on static malware detection in the Literature Review section. The Research Method section describes our main approach to detect malware in Windows environment and explains the process of data collection, feature extraction, and feature selection. We present our empirical results in the Experimental Results section along with the evaluation criteria,

details of different experiments and our findings. We conclude our work in the Conclusion section.

## LITERATURE REVIEW

In this section, we cover previous works done on detecting malware in Windows environment using machine learning methods. Mainly, we will cover some related works in this field which deals with malware detection using static analysis on Windows Portable Executables (PEs).

Several studies have applied machine learning for malware classification and detection. The authors (*Catak, Yazi & Elezaj, 2020*) proposed a long short-term memory (LSTM) method for classifying sequential data of Windows exe API calls. They also contributed to the ongoing research on malware detection by developing a new dataset that contains API calls made on the Windows operating system to represent the behavior of malware. They achieved an accuracy of up to 95%. *Sharma, Rama Krishna & Sahay (2019)* proposed a system based on the frequency of opcode occurrence for detecting malware. The authors used Fisher score, information gain, gain ratio, Chi-square and symmetric uncertainty for selecting top-20 features. They found that five machine learning methods namely random forest, LMT, NBT, J48 Graft and REPTree detect the malware with almost 100% accuracy. *Naval et al. (2015)* focus on proposing an evasion-proof solution that is not vulnerable to system-call injection attacks. They proposed an approach that characterizes program semantics using asymptotic equipartition property to extract information-rich call sequences. These call sequences are further quantified to detect malicious binaries. The results showed that the solution is effective in identifying real malware instances with 95.4% accuracy. *Tang & Qian (2019)* detected malicious code based on the API call sequence. They converted the API call sequence into a characteristic image that can represent the behavior of the malicious code. The convolutional neural network was used to classify the malicious code into nine families and achieved a true positive rate of 99%. The authors (*Raff et al., 2018*) introduced malware detection from raw byte sequences of the entire executable file using neural networks. In this initial work, they discussed many interesting challenges faced in building a neural network for processing raw byte sequences. *Fuyong & Tiezhu (2017)* proposed a new malware detection and classification method based on n-grams attribute similarity. We extract all n-grams of byte codes from training samples and select the most relevant as attributes. After calculating the average value of attributes in malware and benign separately, we determine a test sample is malware or benign by attribute similarity. The results of this study show that the proposed system outperforms traditional machine learning methods. *Wojnowicz et al. (2016)* developed a method to quantify the extent to which patterned variations in a fileâôs entropy signal make it suspicious. By extracting only string and entropy features from samples, they can obtain almost 99% detection of parasitic malware.

*Zhang et al. (2020)* explored function call graph vectorization representation (FCGV) as the input feature to machine learning algorithms for classification and noted that this representation loses some critical features of PE files due to the hash technique being

used. They improved the classification accuracy of the FCGV-based machine learning model by applying both graph and non-graph features and achieved a maximum accuracy of 99.5% with non-graph, *i.e.,* statistical features. The authors (*Chowdhury, Rahman & Islam, 2017*) used the n-gram approach on PE files. First, they extracted PE Header and 5-grams as features and then applied PCA to reduce and focus on the important features only. They achieved an accuracy of 97.7% using an artificial neural network. *Cepeda, Chia Tien & Ordóñez (2016)* found that nine features are enough to distinguish malware from benign files with an accuracy of 99.60%. *Kim et al. (2021)* proposed a static analysis automation technique for detecting malicious code using a portable executable structure. They extracted 12 attributes from 54 attributes of PE structure based on the importance score, however, the system achieved a maximum of 80% accuracy of malicious code classification. The Zero-Day Vigilante (ZeVigilante) system (*Alhaidari et al., 2022*) can detect the malware considering both static and dynamic analyses. They applied six different classifiers and observed that RF achieved the best accuracy for both static and dynamic analyses, 98.21% and 98.92%, respectively. Similarly, the studies (*Patidar & Khandelwal, 2019*; *Gupta & Rani, 2018*; *Kumar & Singh, 2018*; *Venkatraman & Alazab, 2018*) claim zero-day malware detection using machine learning techniques. The study (*Zhang, Kuo & Yang, 2019*) focuses on malware type detection or classification of malware family instead of binary classification. The work uses several machine learning models to build static malware type classifiers on PE-format files. The evaluation results show that random forest can achieve high performance with a micro average F1-score of 0.96 and a macro average F1-score of 0.89. The work (*Pham, Le & Vu, 2018*) proposes a static malware detection method by Portable Executable analysis and Gradient Boosting decision tree algorithm. The method reduces the training time by appropriately reducing the feature dimension and achieves 99.3% accuracy.

*Zhang, Liu & Jiang (2022)* argued that most malware solutions only detect malware families that were included in the training data. They proposed to use a soft relevance value based on multiple trained models. They used features such as file sizes, function call names, DLLs, n-grams, etc. When the models are trained, we try to predict which malware family from the dataset they belong to. By using the trained models, the soft relevance value is applied to find if the malware belongs to one of the original malware families or not. *Singh & Singh (2020)* proposed a behavior-based malware detection technique. Firstly, printable strings are processed word by word using text mining techniques. Secondly, Shannon entropy is computed over the printable strings and API calls to consider the randomness of API and finally, all features are integrated to develop the malware classifiers using the machine learning algorithms. *Cannarile et al. (2022)* presented a benchmark to compare deep learning and shallow learning techniques for API calls malware detection. They considered random forest, CatBoost, XGBoost, and ExtraTrees as shallow learning methods whereas TabNet and NODE (Neural Oblivious Decision Ensembles) were used as deep learning methods. Based on experimental results, they concluded that shallow learning techniques tend to perform better and converge faster(with less training time) to a suitable solution. *Euh et al. (2020)* propose low-dimensional but effective features for a malware detection system and analyze them with tree-based ensemble models. They extract the five

types of malware features represented from binary or disassembly files. The experimental work shows that the tree-based ensemble model is effective and efficient for malware classification concerning training time and generalization performance. *Amer & Zelinka (2020)* introduced the use of word embedding to understand the contextual relationship that exists between API functions in the malware call sequence. Their experimental results prove that there is a significant distinction between malware and goodware call sequences. Next, they introduce a new system to detect and predict malware based on the Markov chain.

In conclusion, there is a vast amount of research on malware detection using machine learning and deep learning. Upon reviewing prior studies, a few key points can be noted. Firstly, most prior works only utilize one or two raw features in their malware detection efforts. Only a limited number of studies have combined raw features to create new ones. Secondly, feature selection is usually done through either Information Gain or principal component analysis, with few studies employing both methods. Thirdly, ensemble learning is not widely used in these studies. Our study, however, extracts four raw features and creates two integrated features. We also apply both Information Gain and principal component analysis for feature selection, use seven different classifiers for malware classification, and incorporate three ensemble learning techniques to increase classification accuracy.

# RESEARCH METHODS

In this section, we discuss our approach to detect PE malware in Windows environment. Our approach can be divided into two phases; malware collection and malware detection.

## Malware collection

We collected the data from MalwareBazaar Database (https://bazaar.abuse.ch/) using its API. The MalwareBazaar Database offers a Comma Separated Values (CSV) file containing the basic information such as SHA256 hash, file name, file type, and signature of all the malware samples available in the database. It also provides an API to download the samples using the information given in the CSV file. We wrote a small script in Python and downloaded more than 30,000 samples of different types of malware. We targeted only PE files in our API calls. The motivation for using PE files was arrived at by monitoring the submissions received over different malware databases. For example, more than 26% malware samples in the malwarebazaar database are PE malware and make it a common file type for spreading malware. Similarly, 47.8% files submitted to Virustotal for analysis are PE files *Kumar, Kuppusamy & Aghila (2019)*. We discarded samples with incorrect values of PE header and samples with code obfuscation *O'Kane, Sezer & McLaughlin (2011)*. After discarding unwanted samples, we have a total of 27,920 samples divided into six categories in our dataset as described in Table 1. We also collected 1,878 benign files from various sources including files from Windows installation. We will make this dataset public very soon.

## Feature sets

We create four feature sets from our data. Moreover, we also create two integrated feature sets by combining these features.

**Table 1  Description of KIET dataset.**

| Malware Type | Count | Description |
|---|---|---|
| RedLineStealer | 5,090 | This is a password stealer type of Spyware. It steals passwords, credit card information and other sensitive data and sends it to a remote location. |
| Downloader | 5,047 | This is a Trojan downloader used by attackers to distribute malware on a large scale. This dataset contains both GuLoader and SmokerLoader samples. |
| RAT | 4,973 | These are Remote Access Trojans (RAT) that allow an attacker to remotely control an infected computer. The samples include AveMariaRAT and njRAT. |
| BankingTrojan | 4,864 | This is a banking Trojan that targets both businesses and consumers for their data, such as banking information, account credentials and bitcoins etc. This dataset contains both TrickBot and QuakBot samples. |
| SnakeKeyLogger | 4,240 | This is a KeyLogger that keeps track of and records victim's keystrokes as s/he types. It is also a spyware and send the recorded information to the hecker through a command and control server. |
| Spyware | 3,706 | This is AgentTesla Spyware that is used by attackers to spy on victims. It can record keystrokes and user interactions on supported programs and web browsers. |
| Total | 27,920 | |
| Benign | 1,877 | Legitimate or goodware files collected from different sources including Windows installation files. |
| Grand Total | 29,797 | |

**Dynamic link libraries:** The first set of features is a list of dynamic link libraries (or DLLs for short) used by each Windows executable. A DLL is a library that contains code, data, and resources that can be used by more than one program at the same time. Windows programs use DLLs to share functionality and resources between themselves. For example, the Comdlg32 DLL performs common dialog box related functions in Windows. A program's certain characteristics can be inferred from the set of DLLs it uses. Therefore, we make a list of DLLs called by malware and benign files to help distinguish between them.

**API functions:** The second set of features is a list of API (Application Program Interface) function names called within the DLLs discovered in the first feature set. Windows APIs are implemented through DLLs and each DLL may contain hundreds of functions in it. A program can be distinguished from others based on the API functions it imports from a DLL. By collecting the list of API functions, we supplement our first feature set in the hope of further improving our ability to differentiate between benign and malware files. The list of API functions can reveal the behavior of the program.

**PE Header:** PE header contains useful information about the executable. A PE file contains a number of headers including MS-DOS Stub, COFF file header, an optional header, etc. They contain metadata about the file itself such as the number of sections, the size of the code, the characteristics of the file, etc. We collect the values of 52 fields of PE

**Table 2** The list of 52 fields of PE Header in our 3rd feature set.

| Header Name | Field Name |
| --- | --- |
| DOS Header | e_magic, e_cblp, e_cp, e_crlc, e_cparhdr, e_minalloc, e_maxalloc, e_ss, e_sp, e_csum, e_ip, e_cs, e_lfarlc, e_ovno, e_oemid, e_oeminfo, e_lfanew |
| File Header | Machine, NumberOfSections, TimeDateStamp, PointerToSymbolTable, NumberOfSymbols, SizeOfOptionalHeader, Characteristics |
| Optional Header | Magic, MajorLinkerVersion, MinorLinkerVersion, SizeOfCode, SizeOfInitializedData, SizeOfUninitializedData, AddressOfEntryPoint, BaseOfCode, ImageBase, SectionAlignment, FileAlignment, MajorOperatingSystemVersion, MinorOperatingSystemVersion, MajorImageVersion, MinorImageVersion, MajorSubsystemVersion, MinorSubsystemVersion, Reserved1, SizeOfImage, SizeOfHeaders, CheckSum, Subsystem, DllCharacteristics, SizeOfStackReserve, SizeOfHeapReserve, SizeOfHeapCommit, LoaderFlags, NumberOfRvaAndSizes |

Header as our third feature set as detailed in Table 2. We get 17 fields from DOS Header, 7 from File Header and 28 from the Optional Header.

**PE sections:** A PE file contains many sections such as executable code section (.text), data sections (.data, .rdata, .bss), and resource section (.rsrc), etc. These sections provide a logical and physical separation of the different parts of a program. Since different programs need different sections depending on their functionality, therefore, collecting information about PE sections could be useful in distinguishing files from each other. Each section in PE has properties such as VirtualAddress, VirtualSize, SizeofRawData, etc. We collect the values of ten properties of each of the ten sections (.text, .data, .rdata, .bss, .idata, .edata, .rsrc, .reloc, .tls, .pdata) as our fourth feature set. In a nutshell, this feature set contains 100 features of the PE section as detailed in Table 3.

**Integrated feature set1 (IFS1):** We combine DLLs referred and API functions called by a sample to create our first integrated feature set. Since both the original feature sets contain names of the DLLs and API functions, therefore, we can simply merge them to create an integrated feature set.

**Integrated feature set2 (IFS2):** We combine the PE header and section feature sets to create our second integrated feature set as both the sets contain numeric values and hence can be merged efficiently.

## Extracting raw feature

In static malware analysis, we can extract useful information from PE files without running the executable. Our PE extractor extracts all the information such as DLLs, API functions, PE Header and Section information and stores them in separate CSV files. We run our PE extractor on all 27,920 malware and 1,877 benign files and store the raw features in four CSV files, one CSV file per feature set.

**Table 3  Ten PE sections and their ten fields in our 4th feature set.**

| Section Name | Description |
| --- | --- |
| .text | This section contains the executable code. It also contains program entry point. |
| .data | This section contains initialized data of a program. |
| .rdata | It contains data that is to be only readable, such as literal strings, and constants. |
| .bss | It represents uninitialized data to reduce the size of executable file. |
| .idata | This section has data about imported functions. |
| .edata | This section contains symbols related information that can be accessed through dynamic linking by other images. |
| .rsrc | This resource-container section contains resource information. |
| .reloc | Relocation information is saved in this section. |
| .tls | TLS stands for Thread Local Storage. Each thread running in Windows uses its own storage called TLS. |
| .pdata | It stores function table entries for exception handling. |
| **Field Name** | **Description** |
| Name | An 8-byte encoded string contains name of the section. |
| Misc_VirtualSize | The total size of the section when loaded into memory. |
| VirtualAddress | The address of the first byte of a section. |
| SizeOfRawData | The size of the section. |
| PointerToRawData | The file pointer to the first page of the section within the COFF file. |
| PointerToRelocations | The file pointer to the beginning of relocation entries for the section. |
| PointerToLinenumbers | The file pointer to the beginning of line-number entries for the section. |
| NumberOfRelocations | The number of relocation entries for the section. |
| NumberOfLinenumbers | The number of line-number entries for the section. |
| Characteristics | The flags that describe the characteristics of the section. |

**DLLs imported:** We pass all the PE files, both malware and benign, to our PE extractor and enlist the names of DLLs called by them. On average, a malware calls four DLLs whereas a benign file calls three DLLs in this study. Overall, malware files import 531 unique DLLs whereas benign files import 186 unique DLLs. The normalized frequency of the top 20 DLLs imported by malware and benign are given in Appendix A. (see Table A1).

To feed DLLs raw features to our classifiers, we apply a Bag-of-Words (BoW) approach for representing the DLLs for each sample. That is, we make a large list of unique names of the DLLs and construct a feature vector for each sample such that each index corresponds to a specific DLL whose value could be either 1 or 0, indicating whether that DLL was called in the file or not. The overall dimensionality, the names of DLLs in the bag, found for our data set was 629.

**API functions:** We extract the names of API functions called within the DLLs extracted above. On average, a malware calls seven API functions whereas a benign calls eight

functions. Overall, malware files call 18,432 unique API functions whereas benign files call 4,256 unique API functions. The normalized frequency of the top 20 API functions called by malware and benign are given in Appendix A (see Table A2).

Similar to the DLLs feature set, we create a Bag-of-Words, *i.e.,* a bag of API functions for constructing a feature vector for each file. The dimensionality of this feature vector was found to be 21,918.

**PE Header:** We extract the values of 17 fields from the DOS header, 7 from the COFF file header, and 28 from the optional header; a total of 52 fields from the PE header. Since these are numeric values, therefore, we normalize them and create a 52 dimensional vector for each file to represent this feature set.

**PE section:** We extract the values of ten fields each from ten sections of PE. (see Table 3). All ten fields are numeric except the Name field so we omit this field and use the normalized values of the other nine fields from each section. This gives us a feature vector of 90 dimensions.

**Integrated feature set1 (IFS1):** We simply merge the first two feature sets, i.e, DLLs imported and API Functions to form this integrated feature set. The integrated feature vector contains 22,575 features in total.

**Integrated feature set2 (IFS2):** We form this integrated feature set by merging 52 fields of the PE header and 90 fields of the PE section. The integrated feature vector has 142 dimensions.

## Feature selection

The raw features are numerous especially in the case of DLLs imported and API functions called by a sample and it is possible that some features might not contain useful information for a machine learning model. Therefore, we applied two feature selection or dimensionality reduction techniques namely Information Gain (IG), and principal component analysis (PCA). By applying these feature selection techniques, we are able to reduce the number of features in each feature set significantly. As a result, we decrease the processing time to train and test the classifiers and possibly also improve their accuracy in detecting the malware. The total number of features in raw feature sets and selected feature sets after applying Information Gain and principal component analysis are given in Table 4.

## EXPERIMENTAL RESULTS

Our dataset consists of a total of 29,797 PE samples, of them 27,920 are malware and 1,877 are benign or goodware. We apply seven machine learning classifiers, two feature selection techniques, and three ensemble learning methods to detect malware as depicted in Fig. 1. We use standard 10-fold cross-validation for training and testing our models. It means we randomly divide our dataset into 10 smaller subsets such that nine subsets are used for training and 1 subset is used for testing. We repeat this process 10 times for every combination. This methodology helps evaluate the robustness of any approach to detect malware without any a priori information. Moreover, the dataset was split into 70:30 ratio for training and testing purposes respectively, *i.e.,* 70% data is used for training the classifiers whereas 30% is used for testing them.

**Table 4** The number of features in raw feature sets and selected feature sets after applying Information Gain and principal component analysis.

| Feature Type | Raw Features | After PCA | After IG |
| --- | --- | --- | --- |
| DLLs Imported | 629 | 195 | 125 |
| API Functions | 21,918 | 494 | 219 |
| PE Header | 52 | 19 | 10 |
| PE Section | 90 | 26 | 18 |
| Integrated Feature Set1 | 22,547 | 563 | 225 |
| integrated Feature Set2 | 142 | 49 | 28 |

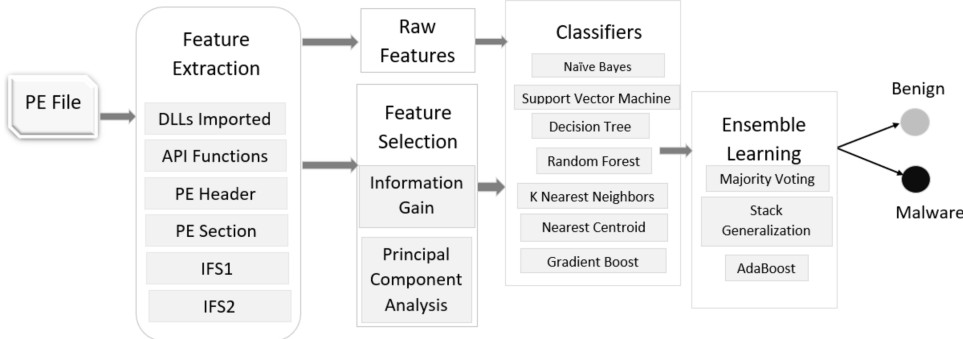

**Figure 1** Malware detection phase.

## Evaluation criteria

To evaluate the performance of our system, we create a confusion matrix for each classifier. A confusion matrix summarizes the performance of a classifier in the form of a table with the help of four quantities namely True Positive (TP), True Negative (TN), False Positive (FP), and False Negative (FN). It helps in measuring the accuracy, recall, precision, and F-score of a classifier. We briefly define the metrics we used to measure the performance of our system.

**True Positive (TP):** A malware classified as a malware.

**True Negative (TN):** A benign classified as a benign.

**False Positive (FP):** A benign classified as a malware.

**False Negative (FN):** A malware classified as a benign.

**Accuracy (ACC):** It calculates the correctly classified or predicted samples by the system as the ratio of correct predictions to the total predictions.

$$ACC = \frac{TP + TN}{TP + TN + FP + FN}$$

**Error rate (ERR):** It is calculated as the ratio of the number of incorrect predictions to the total predictions. It is also called misclassification.

$$ERR = \frac{FP + FN}{TP + TN + FP + FN}$$

**Recall**: The recall is calculated as the number of correct positive predictions divided by the total number of positives. It is also called True Positive Rate (TPR).

$$Recall = \frac{TP}{TP + FN}$$

**Precision**: The precision is calculated as the number of correct positive predictions divided by the total number of positive predictions.

$$Precision = \frac{TP}{TP + FP}$$

**F-score** It is a harmonic mean of precision and recall.

$$F-score = 2 * \frac{Precision * Recall}{Precision + Recall}.$$

## Experimental setup

To validate the proposed system shown in Fig. 1, we create an experimental setup on the Windows operating system running on AMD Ryzen 7 4800H @4.2 GHz processor and 16 GB of main memory. We use the Scikit-learn (*Pedregosa et al., 2011*) library of Python to run all the experiments. It has the implementation of many classifiers and helps in splitting the data into training and testing, 10-fold cross-validation, and comparing the performance of different classifiers using confusion matrix and other metrics.

## Testing with raw features

In the first experiment, we evaluate our system on raw features. We apply all the classifiers and ensemble learning methods on individual and integrated features, and the results are presented in Table 5. We see that in the case of imported DLLs, the random forest model outperforms other classifiers with an accuracy of 96.41% and an error rate of only 3.59%. The Stack Generalization gives the best accuracy of 96.47% on this feature set. We see the same trend in other feature sets. The random forest model achieves the best accuracy of 99.36% on the PE Header feature set while the worst performance with 92.0% accuracy is attained by the Nearest Centroid method on the PE Section feature set.

Regarding the integrated features, the decision tree model and AdaBoost ensemble learning method outperform with accuracies of 97.69% and 97.85% respectively when we integrate DLLs and API functions into one feature set, i.e, IFS1. For combined PE Header and Section, i.e, IFS2, random forest, and Stack Generalization give the best results with accuracies of 99.41% and 99.48% respectively whereas naive Bayes gives the worst results. On average, we detect malware with an accuracy of more than 97% using raw features. The maximum F-score of 0.997 is delivered by both the random forest model and Stack Generalization method on IFS2 while the minimum error rate is given by Stack Generalization on the same feature set.

## Testing with selected features

In the second experiment, we evaluate the performance of our system on selected features after applying Information Gain (IG) and principal component analysis (PCA) methods to

**Table 5  The performance of different classifiers and ensemble learning techniques on individual and integrated raw features.**

| Feature | Classifier | ACC (%) | ERR (%) | Recall | Precision | F-Score |
|---|---|---|---|---|---|---|
| DLLs Imported | Naïve Bayes | 96.03 | 3.97 | 0.998 | 0.961 | 0.979 |
| | SVM | 96.27 | 3.73 | 0.998 | 0.964 | 0.981 |
| | Decision Tree | 96.37 | 3.63 | 0.998 | 0.965 | 0.981 |
| | Random Forest | 96.41 | 3.59 | 1.000 | 0.964 | 0.981 |
| | KNN | 96.20 | 3.80 | 0.999 | 0.962 | 0.980 |
| | Nearest Centroid | 95.73 | 4.27 | 0.996 | 0.960 | 0.978 |
| | Gradient Boost | 96.08 | 3.92 | 1.000 | 0.960 | 0.980 |
| | Ensemble Learning | | | | | |
| | Majority Voting | 96.31 | 3.69 | 0.999 | 0.963 | 0.981 |
| | Stacking Generalization | 96.47 | 3.53 | 0.999 | 0.965 | 0.982 |
| | AdaBoost | 96.41 | 3.59 | 0.998 | 0.965 | 0.981 |
| API Functions | Naïve Bayes | 95.42 | 4.58 | 0.988 | 0.965 | 0.976 |
| | SVM | 94.84 | 5.15 | 1.000 | 0.948 | 0.973 |
| | Decision Tree | 96.46 | 3.54 | 0.999 | 0.965 | 0.982 |
| | Random Forest | 96.59 | 3.41 | 0.999 | 0.966 | 0.982 |
| | KNN | 95.56 | 4.44 | 0.999 | 0.955 | 0.977 |
| | Nearest Centroid | 95.80 | 4.20 | 0.994 | 0.963 | 0.978 |
| | Gradient Boost | 96.51 | 3.49 | 0.999 | 0.965 | 0.982 |
| | Ensemble Learning | | | | | |
| | Majority Voting | 96.19 | 3.81 | 1.000 | 0.961 | 0.980 |
| | Stacking Generalization | 96.37 | 3.63 | 0.999 | 0.964 | 0.981 |
| | AdaBoost | 96.36 | 3.64 | 0.999 | 0.963 | 0.981 |
| PE Header | Naïve Bayes | 95.09 | 4.91 | 0.994 | 0.956 | 0.974 |
| | SVM | 97.10 | 2.90 | 0.995 | 0.975 | 0.985 |
| | Decision Tree | 99.11 | 0.89 | 0.997 | 0.994 | 0.995 |
| | Random Forest | 99.36 | 0.64 | 0.999 | 0.994 | 0.997 |
| | KNN | 98.71 | 1.29 | 0.994 | 0.992 | 0.993 |
| | Nearest Centroid | 93.74 | 6.26 | 0.978 | 0.957 | 0.967 |
| | Gradient Boost | 98.86 | 1.14 | 0.997 | 0.991 | 0.994 |
| | Ensemble Learning | | | | | |
| | Majority Voting | 98.83 | 1.17 | 0.998 | 0.989 | 0.994 |
| | Stacking Generalization | 99.31 | 0.69 | 0.998 | 0.994 | 0.996 |
| | AdaBoost | 99.11 | 0.89 | 0.997 | 0.994 | 0.995 |

**Table 5** (*continued*)

| Feature | Classifier | ACC (%) | ERR (%) | Recall | Precision | F-Score |
|---|---|---|---|---|---|---|
| PE Section | Naïve Bayes | 94.02 | 5.98 | 0.989 | 0.949 | 0.969 |
| | SVM | 95.80 | 4.20 | 0.996 | 0.961 | 0.978 |
| | Decision Tree | 96.47 | 3.53 | 0.981 | 0.981 | 0.981 |
| | Random Forest | 97.32 | 2.67 | 0.991 | 0.981 | 0.986 |
| | KNN | 96.72 | 3.28 | 0.988 | 0.978 | 0.983 |
| | Nearest Centroid | 92.00 | 8.00 | 0.956 | 0.959 | 0.958 |
| | Gradient Boost | 97.01 | 2.99 | 0.996 | 0.973 | 0.984 |
| | Ensemble Learning | | | | | |
| | Majority Voting | 97.03 | 2.97 | 0.998 | 0.972 | 0.984 |
| | Stacking Generalization | 97.31 | 2.70 | 0.992 | 0.980 | 0.986 |
| | AdaBoost | 96.69 | 3.31 | 0.984 | 0.981 | 0.982 |
| Integrated Feature Set1 | Naïve Bayes | 95.97 | 4.03 | 0.988 | 0.970 | 0.979 |
| | SVM | 95.69 | 4.31 | 1.000 | 0.956 | 0.978 |
| | Decision Tree | 97.69 | 2.31 | 0.999 | 0.977 | 0.988 |
| | Random Forest | 97.53 | 2.47 | 0.999 | 0.975 | 0.987 |
| | KNN | 95.80 | 4.20 | 0.999 | 0.958 | 0.978 |
| | Nearest Centroid | 95.64 | 4.36 | 0.993 | 0.962 | 0.977 |
| | Gradient Boost | 97.24 | 2.76 | 1.000 | 0.972 | 0.986 |
| | Ensemble Learning | | | | | |
| | Majority Voting | 97.05 | 2.95 | 1.000 | 0.970 | 0.985 |
| | Stacking Generalization | 97.68 | 2.32 | 0.998 | 0.977 | 0.988 |
| | AdaBoost | 97.85 | 2.15 | 0.999 | 0.979 | 0.989 |
| Integrated Feature Set2 | Naïve Bayes | 93.23 | 6.77 | 0.973 | 0.956 | 0.964 |
| | SVM | 97.61 | 2.39 | 0.998 | 0.977 | 0.987 |
| | Decision Tree | 99.14 | 0.86 | 0.996 | 0.995 | 0.995 |
| | Random Forest | 99.41 | 0.59 | 0.999 | 0.995 | 0.997 |
| | KNN | 98.52 | 1.48 | 0.994 | 0.991 | 0.992 |
| | Nearest Centroid | 93.67 | 6.33 | 0.974 | 0.960 | 0.967 |
| | Gradient Boost | 98.91 | 1.09 | 0.998 | 0.991 | 0.994 |
| | Ensemble Learning | | | | | |
| | Majority Voting | 98.87 | 1.13 | 0.999 | 0.989 | 0.994 |
| | Stacking Generalization | 99.48 | 0.52 | 0.999 | 0.996 | 0.997 |
| | AdaBoost | 99.14 | 0.86 | 0.996 | 0.995 | 0.995 |

choose important features. The results obtained with the features selected using Information Gain are presented in Table 6. The table shows that the performance of different classifiers slightly decreases when compared to their performance on raw features. However, overall the performance improves on integrated feature sets. Moreover, using selected features we can lessen the training time significantly. We achieve the best accuracy of 99.5% and the best F-score of 0.998 with the Stack Generalization method on IFS2 when we apply it to the top 20% features ranked by their IG score. Similarly, the results shown in Table 7 depict that the contribution of PCA transformation slimly deteriorates the performance. We achieve the best accuracy of 99.41% and the best F-score of 0.997 with the Stack Generalization method on IFS2 when we apply it to the top 30% features selected by their

principal components. The reason that the accuracy does not improve significantly with selected features is that, on average, we select only 20% and 30% most important features after applying IG and PCA respectively (In the case of API Functions we use the top 1% features). The main purpose of selecting a small number of important features was to reduce the training time significantly while maintaining a good overall performance. On the other side, the ensemble learning techniques show promising results on both the raw and selected features and their performance improves marginally on selected integrated features.

## ROC curves

Receiver operating characteristic (ROC) curve graphically shows the performance of a classifier at all classification thresholds. It is created by plotting Recall or True Positive Rate (TPR) against Specificity or False Positive Rate (FPR) where Specificity is calculated as $\frac{TN}{TN+FP}$. ROC curve depicts the discriminative ability of a binary classifier and is considered a good metric when class imbalance might lead to accuracy paradox (*Gibert, Mateu & Planes, 2020*).

Figure 2 shows the ROC curves for six classifiers on raw feature sets. It excludes the NC classifier as we cannot compute probabilities in NC. The figure also shows the AUC or Area Under the Curve for each ROC. Both ROC and AUC values confirm that all the classifiers give a good performance because the feature sets help them in discriminating different classes of malware at all thresholds. The figure also shows that the AUC values of IFS2 are very promising and reach the maximum values of 1.00 and 0.99 for RF and GB classifiers respectively whereas these values are 1.00 and 0.98 with the PE Header feature set. We can see similar trends for other classifiers where the integrated feature sets improve the discriminating ability of a classifier as its AUC values increase. Random Forest outperforms all other classifiers on both individual and integrated features. The ROC curves on selected features after applying IG and PCA are given in Appendix A (See Figs. A1 and A2).

## 10-fold cross validation

As mentioned above, we use 10-fold cross-validation for training and testing the models. Since a single train-test split has limitations such as the split might not represent each class proportionally, therefore, the roust cross-validation method is becoming a default. For 10-fold cross-validation, we split the dataset into 10 folds and for 10 times the 9 folds are used for training and the one fold is used for testing. The final result is given as the average of all 10 folds.

Figure 3 shows the accuracy of each model during 10-fold cross-validation in the form of a box plot for each classifier. The figure shows that there is more variation in the case of DLLs Imported and API Functions feature sets. Though the variation decreases when we combine these features in IFS1, it is still more than that of other features. A possible reason is that in every fold the feature vectors had very different words from the bag of words and the sparseness of these vectors gave diverse results. We can also see that there are more outliers in this case compared to other feature sets. On the other hand, PE Header, PE Section, and IFS2 have low variation as their feature vectors have normalized numeric

**Table 6** The performance of different classifiers and ensemble learning techniques on individual and integrated features selected using the Information Gain method.

| Feature | Classifier | ACC (%) | ERR (%) | Recall | Precision | F-Score |
|---------|-----------|---------|---------|--------|-----------|---------|
| DLLs Imported | Naïve Bayes | 95.36 | 4.64 | 0.997 | 0.956 | 0.976 |
| | SVM | 95.80 | 4.20 | 0.999 | 0.958 | 0.978 |
| | Decision Tree | 95.83 | 4.17 | 0.999 | 0.959 | 0.978 |
| | Random Forest | 95.78 | 4.22 | 0.999 | 0.958 | 0.978 |
| | KNN | 95.78 | 4.22 | 0.999 | 0.958 | 0.978 |
| | Nearest Centroid | 95.49 | 4.51 | 0.997 | 0.957 | 0.977 |
| | Gradient Boost | 95.73 | 4.27 | 1.000 | 0.957 | 0.978 |
| | Ensemble Learning | | | | | |
| | Majority Voting | 96.31 | 3.69 | 1.000 | 0.963 | 0.981 |
| | Stacking Generalization | 96.46 | 3.54 | 0.998 | 0.965 | 0.982 |
| | AdaBoost | 96.47 | 3.53 | 0.999 | 0.965 | 0.982 |
| API Functions | Naïve Bayes | 93.65 | 6.35 | 0.969 | 0.964 | 0.966 |
| | SVM | 96.58 | 3.42 | 1.000 | 0.965 | 0.982 |
| | Decision Tree | 96.37 | 3.63 | 0.997 | 0.965 | 0.981 |
| | Random Forest | 96.48 | 3.52 | 0.998 | 0.965 | 0.982 |
| | KNN | 96.39 | 3.61 | 0.998 | 0.965 | 0.981 |
| | Nearest Centroid | 94.41 | 5.59 | 0.978 | 0.963 | 0.971 |
| | Gradient Boost | 96.49 | 3.51 | 1.000 | 0.964 | 0.982 |
| | Ensemble Learning | | | | | |
| | Majority Voting | 96.58 | 3.42 | 1.000 | 0.965 | 0.982 |
| | Stacking Generalization | 96.42 | 3.58 | 0.998 | 0.965 | 0.981 |
| | AdaBoost | 96.37 | 3.63 | 0.998 | 0.965 | 0.981 |
| PE Header | Naïve Bayes | 94.21 | 5.79 | 1.000 | 0.942 | 0.970 |
| | SVM | 95.54 | 4.46 | 0.995 | 0.959 | 0.977 |
| | Decision Tree | 98.76 | 1.24 | 0.995 | 0.992 | 0.993 |
| | Random Forest | 99.11 | 0.89 | 0.997 | 0.993 | 0.995 |
| | KNN | 98.59 | 1.41 | 0.995 | 0.990 | 0.993 |
| | Nearest Centroid | 94.75 | 5.25 | 0.988 | 0.958 | 0.973 |
| | Gradient Boost | 98.64 | 1.36 | 0.996 | 0.990 | 0.993 |
| | Ensemble Learning | | | | | |
| | Majority Voting | 98.78 | 1.22 | 0.998 | 0.989 | 0.994 |
| | Stacking Generalization | 99.31 | 0.70 | 0.998 | 0.994 | 0.996 |
| | AdaBoost | 99.08 | 0.92 | 0.997 | 0.993 | 0.995 |

**Table 6** (*continued*)

| Feature | Classifier | ACC (%) | ERR (%) | Recall | Precision | F-Score |
|---|---|---|---|---|---|---|
| PE Section | Naïve Bayes | 94.19 | 5.81 | 1.000 | 0.942 | 0.970 |
| | SVM | 95.41 | 4.59 | 0.997 | 0.956 | 0.976 |
| | Decision Tree | 96.14 | 3.86 | 0.980 | 0.979 | 0.980 |
| | Random Forest | 96.89 | 3.11 | 0.991 | 0.976 | 0.984 |
| | KNN | 96.44 | 3.56 | 0.989 | 0.973 | 0.981 |
| | Nearest Centroid | 92.89 | 7.11 | 0.967 | 0.958 | 0.962 |
| | Gradient Boost | 96.66 | 3.34 | 0.996 | 0.969 | 0.982 |
| | Ensemble Learning | | | | | |
| | Majority Voting | 96.98 | 3.02 | 0.997 | 0.971 | 0.984 |
| | Stacking Generalization | 97.33 | 2.67 | 0.993 | 0.979 | 0.986 |
| | AdaBoost | 96.49 | 3.51 | 0.982 | 0.981 | 0.981 |
| Integrated Feature Set1 | Naïve Bayes | 94.37 | 5.63 | 0.971 | 0.969 | 0.970 |
| | SVM | 97.03 | 2.97 | 0.997 | 0.972 | 0.984 |
| | Decision Tree | 96.97 | 3.07 | 0.997 | 0.971 | 0.984 |
| | Random Forest | 97.05 | 2.92 | 0.997 | 0.973 | 0.985 |
| | KNN | 96.92 | 3.08 | 0.997 | 0.971 | 0.984 |
| | Nearest Centroid | 94.37 | 5.63 | 0.977 | 0.964 | 0.970 |
| | Gradient Boost | 96.95 | 3.05 | 0.997 | 0.971 | 0.984 |
| | Ensemble Learning | | | | | |
| | Majority Voting | 97.02 | 2.98 | 0.997 | 0.972 | 0.984 |
| | Stacking Generalization | 97.07 | 2.92 | 0.997 | 0.973 | 0.985 |
| | AdaBoost | 97.07 | 2.93 | 0.997 | 0.973 | 0.985 |
| Integrated Feature Set2 | Naïve Bayes | 95.35 | 4.65 | 0.996 | 0.956 | 0.976 |
| | SVM | 95.73 | 4.27 | 0.998 | 0.959 | 0.978 |
| | Decision Tree | 99.13 | 0.86 | 0.995 | 0.996 | 0.995 |
| | Random Forest | 99.36 | 0.60 | 0.999 | 0.994 | 0.997 |
| | KNN | 98.20 | 1.80 | 0.994 | 0.987 | 0.990 |
| | Nearest Centroid | 94.81 | 5.19 | 0.991 | 0.956 | 0.973 |
| | Gradient Boost | 98.99 | 1.01 | 0.997 | 0.992 | 0.995 |
| | Ensemble Learning | | | | | |
| | Majority Voting | 98.56 | 1.48 | 0.999 | 0.986 | 0.992 |
| | Stacking Generalization | 99.50 | 0.47 | 0.999 | 0.996 | 0.998 |
| | AdaBoost | 99.16 | 0.76 | 0.996 | 0.996 | 0.996 |

values and there is no chance of having a sparse vector in training or testing. In a nutshell, all the classifiers give high accuracy on all the features but there is more variation in IFS1 and its components compared to IFS2 and its components. The accuracy of different classifiers on selected features after applying PCA and IG with 10-fold cross validation is given in Appendix A. (see Fig. A3 and Fig. A4).

## Performance on raw *vs.* selected features

Table 8 summarizes the maximum accuracy achieved by our system on raw and selected features. Our system is composed of seven classifiers and three ensemble learning techniques as shown in Fig. 1 and we pick the best results in each case for comparison. Table 8 shows

**Table 7** The performance of different classifiers and ensemble learning techniques on individual and integrated features selected using principal component analysis method.

| Feature | Classifier | ACC (%) | ERR (%) | Recall | Precision | F-Score |
|---|---|---|---|---|---|---|
| DLLs Imported | Naïve Bayes | 95.47 | 4.53 | 0.999 | 0.955 | 0.977 |
| | SVM | 95.73 | 4.27 | 0.999 | 0.957 | 0.978 |
| | Decision Tree | 95.64 | 4.36 | 1.000 | 0.956 | 0.977 |
| | Random Forest | 95.61 | 4.39 | 1.000 | 0.956 | 0.977 |
| | KNN | 95.44 | 4.56 | 0.999 | 0.954 | 0.976 |
| | Nearest Centroid | 95.54 | 4.46 | 0.997 | 0.957 | 0.977 |
| | Gradient Boost | 95.39 | 4.61 | 1.000 | 0.954 | 0.976 |
| | Ensemble Learning | | | | | |
| | Majority Voting | 95.66 | 4.34 | 1.000 | 0.956 | 0.977 |
| | Stacking Generalization | 95.69 | 4.31 | 0.999 | 0.957 | 0.978 |
| | AdaBoost | 95.71 | 4.29 | 1.000 | 0.957 | 0.978 |
| API Functions | Naïve Bayes | 94.75 | 5.25 | 0.999 | 0.948 | 0.973 |
| | SVM | 94.70 | 5.30 | 0.998 | 0.948 | 0.973 |
| | Decision Tree | 94.80 | 5.20 | 0.999 | 0.948 | 0.973 |
| | Random Forest | 94.78 | 5.22 | 0.999 | 0.948 | 0.973 |
| | KNN | 94.53 | 5.47 | 1.000 | 0.945 | 0.972 |
| | Nearest Centroid | 94.56 | 5.44 | 0.994 | 0.951 | 0.972 |
| | Gradient Boost | 94.42 | 5.58 | 1.000 | 0.944 | 0.971 |
| | Ensemble Learning | | | | | |
| | Majority Voting | 94.81 | 5.20 | 1.000 | 0.948 | 0.973 |
| | Stacking Generalization | 94.76 | 5.24 | 0.996 | 0.951 | 0.973 |
| | AdaBoost | 94.81 | 5.19 | 0.999 | 0.948 | 0.973 |
| PE Header | Naïve Bayes | 94.21 | 5.79 | 1.000 | 0.942 | 0.970 |
| | SVM | 96.09 | 3.91 | 0.995 | 0.964 | 0.980 |
| | Decision Tree | 98.66 | 1.34 | 0.994 | 0.992 | 0.993 |
| | Random Forest | 99.20 | 0.81 | 0.999 | 0.993 | 0.996 |
| | KNN | 98.54 | 1.46 | 0.994 | 0.991 | 0.992 |
| | Nearest Centroid | 90.81 | 9.19 | 0.936 | 0.965 | 0.950 |
| | Gradient Boost | 98.39 | 1.61 | 0.996 | 0.987 | 0.991 |
| | Ensemble Learning | | | | | |
| | Majority Voting | 98.46 | 1.54 | 0.999 | 0.985 | 0.992 |
| | Stacking Generalization | 99.26 | 0.74 | 0.999 | 0.993 | 0.996 |
| | AdaBoost | 99.11 | 0.89 | 0.998 | 0.992 | 0.995 |

**Table 7** (*continued*)

| Feature | Classifier | ACC (%) | ERR (%) | Recall | Precision | F-Score |
|---|---|---|---|---|---|---|
| PE Section | Naïve Bayes | 94.22 | 5.78 | 1.000 | 0.942 | 0.970 |
| | SVM | 95.70 | 4.30 | 0.996 | 0.960 | 0.978 |
| | Decision Tree | 96.62 | 3.38 | 0.981 | 0.983 | 0.982 |
| | Random Forest | 97.63 | 2.37 | 0.993 | 0.983 | 0.987 |
| | KNN | 96.77 | 3.23 | 0.988 | 0.978 | 0.983 |
| | Nearest Centroid | 92.89 | 7.11 | 0.967 | 0.958 | 0.962 |
| | Gradient Boost | 96.84 | 3.16 | 0.997 | 0.970 | 0.983 |
| | Ensemble Learning | | | | | |
| | Majority Voting | 96.89 | 3.11 | 0.998 | 0.970 | 0.984 |
| | Stacking Generalization | 97.60 | 2.40 | 0.993 | 0.982 | 0.987 |
| | AdaBoost | 96.91 | 3.09 | 0.986 | 0.981 | 0.984 |
| Integrated Feature Set1 | Naïve Bayes | 94.92 | 5.08 | 0.998 | 0.950 | 0.974 |
| | SVM | 94.97 | 5.03 | 0.998 | 0.951 | 0.974 |
| | Decision Tree | 94.95 | 5.05 | 0.999 | 0.950 | 0.974 |
| | Random Forest | 94.93 | 5.07 | 0.999 | 0.949 | 0.974 |
| | KNN | 94.68 | 5.32 | 0.999 | 0.948 | 0.972 |
| | Nearest Centroid | 94.64 | 5.36 | 0.992 | 0.953 | 0.972 |
| | Gradient Boost | 94.49 | 5.51 | 1.000 | 0.945 | 0.972 |
| | Ensemble Learning | | | | | |
| | Majority Voting | 94.95 | 5.05 | 0.999 | 0.950 | 0.974 |
| | Stacking Generalization | 94.93 | 5.07 | 0.998 | 0.950 | 0.974 |
| | AdaBoost | 95.05 | 4.95 | 0.999 | 0.951 | 0.974 |
| Integrated Feature Set2 | Naïve Bayes | 94.28 | 5.76 | 1.000 | 0.942 | 0.970 |
| | SVM | 96.47 | 3.53 | 0.997 | 0.966 | 0.982 |
| | Decision Tree | 98.98 | 1.02 | 0.995 | 0.994 | 0.995 |
| | Random Forest | 99.31 | 0.69 | 0.999 | 0.993 | 0.996 |
| | KNN | 98.66 | 1.34 | 0.996 | 0.990 | 0.993 |
| | Nearest Centroid | 96.11 | 3.89 | 0.988 | 0.962 | 0.953 |
| | Gradient Boost | 98.92 | 1.08 | 0.997 | 0.992 | 0.994 |
| | Ensemble Learning | | | | | |
| | Majority Voting | 98.94 | 1.06 | 1.000 | 0.989 | 0.994 |
| | Stacking Generalization | 99.41 | 0.59 | 0.999 | 0.995 | 0.997 |
| | AdaBoost | 99.16 | 0.84 | 0.996 | 0.995 | 0.996 |

that IFS2 gives the best results on both raw and selected features. It is also clear from the table that our system achieves the best accuracy of 99.50% on this set after selecting important features using the Information Gain method. The table also highlights a very interesting pattern. When we apply IG to select features, the accuracy of our system slightly improves on IFS2 while it marginally decreases on IFS1. On the other side, PCA does not seem to play its role and the accuracy of our system slightly decreases on both raw and selected features. The table also shows that the system performs consistently better on the PE Header feature set than other feature sets. It seems that the PE Header feature set alone or integrated with another feature set (*e.g.*, PE Section) is a good candidate for developing a malware detection system for filtering zero-day malware.

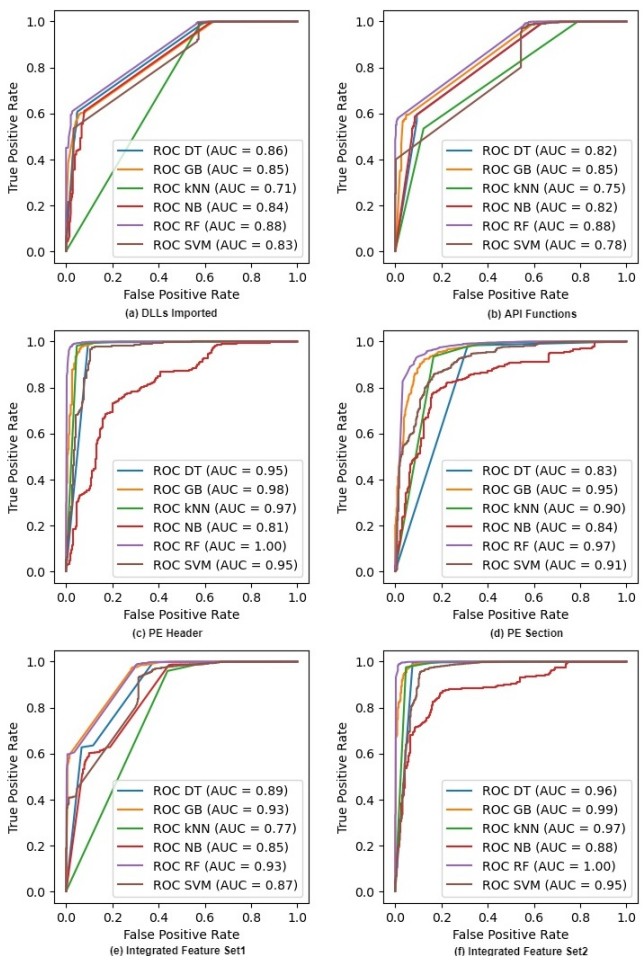

**Figure 2  ROC curves for different classifiers on raw feature sets.**

## Comparison with previous work

In this section, we compare the performance of our system with previous systems or studies to detect malware. We compare our work with some recent works that also applied static malware analysis to detect malware using classic machine learning models. We reproduce these works so that we have a fair comparison. There are more works related to this proposed work. However, either they used different feature sets such as byte-n-gram, opcode-n-gram or they applied deep learning neural networks to detect malware, therefore, we skipped them as they are less relevant.

Table 9 summarizes the accuracy and error rate of the proposed work and the previous works. The work presented by *Kumar, Kuppusamy & Aghila (2019)* uses 53 field values of the PE header as raw features and then creates an integrated feature set having 68 features. The work then applies six classification models on both raw and integrated features. Random Forest model achieves the maximum accuracy of 98.4% and an error rate of 1.47% on integrated features in this work. *Azmee et al. (2020)* extract 77 features of the PE header and deploy nine classifiers to classify malware samples. They achieve the

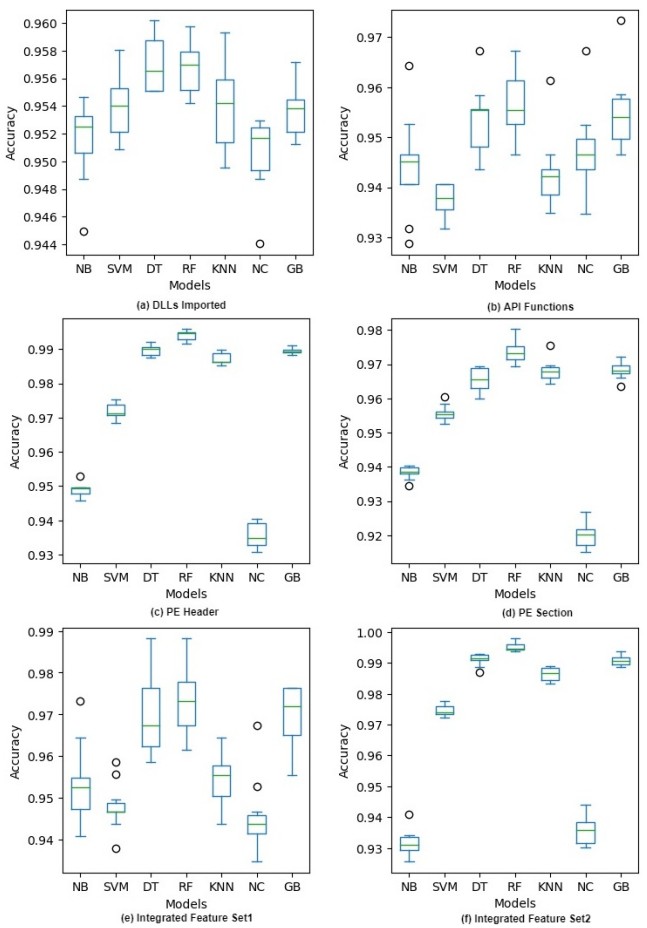

**Figure 3** Accuracy of different classifiers on raw feature sets with 10-fold cross validation.

**Table 8** The maximum accuracy (in percentage) achieved by our system on raw feature sets and selected feature sets after applying Information Gain and principal component analysis.

| Feature type | Raw features | After PCA | After IG |
|---|---|---|---|
| DLLs Imported | 96.47 | 95.73 | 96.47 |
| API Functions | 96.59 | 94.81 | 96.58 |
| PE Header | 99.36 | 99.26 | 99.31 |
| PE Section | 97.32 | 97.63 | 97.33 |
| Integrated Feature Set1 | 97.85 | 95.05 | 97.07 |
| Integrated Feature Set2 | 99.48 | 99.41 | 99.50 |

maximum accuracy and minimum error rate of 98.6% and 1.41% respectively with the XGBoost model. The work *Damaševičius et al. (2021)* implements both machine learning and deep learning models for Windows PE malware detection. ExtraTrees model achieves an accuracy of 98.7% and an error rate of 1.32% on the 68-dimensional feature set of the PE header in this work. *Kim et al. (2021)* first extract 54 attributes from the PE structure and then use the top 12 most important features to classify malware. Their work achieves

**Table 9  Comparison of the proposed work with the previous works.**

| Work | Accuracy | Error rate | Classifier | Feature set |
|---|---|---|---|---|
| **Proposed work** | **99.5%** | **0.47%** | **Random forest** | **Integrated feature set** |
| *Kumar, Kuppusamy & Aghila (2019)* | 98.3% | 1.47% | Random forest | Integrated feature set |
| *Azmee et al. (2020)* | 98.6% | 1.41% | XGBoost | 77 features of PE header |
| *Damaševičius et al. (2021)* | 98.7% | 1.32% | ExtraTrees | 68 features of PE header |
| | 98.7% | 1.31% | AdaBoost | PE Structure |

a maximum accuracy of 98.7% and a minimum error rate of 1.31% using the AdaBoost model. It is clear from Table 9 that our proposed work with an accuracy of 99.5% and error rate of only 0.47% on integrated feature set outperforms previous works. The table shows that the proposed system produces a very small error. In other words, the probability of misclassification in the proposed system is much lower than in the previous systems. We agree that in terms of accuracy the system improvement is marginal, however, when combined with other metrics, the proposed system gives better results, especially in terms of a very small error rate.

# CONCLUSION

The work presents a static malware detection system based on mining DLLs, and API calls from each DLL, PE Header, and PE Section and also combines the features to create integrated features. A new dataset of a total of 27,920 PE samples is collected and the features are extracted to feed them to seven machine learning models and three ensemble learning techniques. Moreover, Information Gain and principal component analysis are used to finding a reduced set of features. The empirical results show that random forest outperforms all other classifiers while decision tree stands second. An accuracy of 99.5% with an error rate of only 0.47% is achieved on the integrated feature set, a combination of PE Header and PE Section. On average, the system's accuracy is greater than 96% while the error rate is below 3.5%. The feature set having the values of PE Header turns out to be the best feature set and when combined with PE Section, the resulting integrated feature set gives the maximum accuracy. Furthermore, the system surpasses the previous studies in terms of higher accuracy and lower error rate.

As a tangible outcome, a preprocessed dataset having 27,920 malware samples is created and available on request along with raw and integrated feature sets for comparing future work with the proposed work. We tested the proposed malware detection system extensively and performed multiple experiments on raw and integrated features to check its performance. By applying two feature selection methods, seven machine learning classifiers and three ensemble learning techniques on multiple features, we tried to bridge the gap in the previous works on malware detection.

Our experiments show that PE Header forms the best feature set and gives the maximum accuracy and minimum error rate when integrated with PE Section. However, real-world

scenario can be different from the experimental environment, hence, we cannot recommend using PE Header alone to detect malware. But, we do assert that this could be a starting point to further explore PE Header and its fields to develop a feature set for detecting zero-day attacks accurately and quickly. In future studies, we can add more file formats such as image, pdf, audio, video, etc. We can also work on adding mobile environments such as iOS and android.

## APPENDIX A

**Table A1  The normalized frequency of top 20 DLLs imported by malware and benign files.**

| No. | Malware | | Benign | |
| | DLL | Normalized frequency | DLL | Normalized frequency |
| --- | --- | --- | --- | --- |
| 1 | kernel32.dll | 0.1782 | mscoree.dll | 0.2041 |
| 2 | mscoree.dll | 0.1465 | kernel32.dll | 0.1589 |
| 3 | user32.dll | 0.1399 | msvcrt.dll | 0.0911 |
| 4 | gdi32.dll | 0.0978 | user32.dll | 0.0653 |
| 5 | advapi32.dll | 0.0820 | advapi32.dll | 0.0552 |
| 6 | comctl32.dll | 0.0468 | ole32.dll | 0.0369 |
| 7 | msvbvm60.dll | 0.0406 | libintl-8.dll | 0.0357 |
| 8 | shell32.dll | 0.0403 | libglib-2.0-0.dll | 0.0351 |
| 9 | oleaut32.dll | 0.0386 | shell32.dll | 0.0343 |
| 10 | ole32.dll | 0.0376 | libgimp-2.0-0.dll | 0.0335 |
| 11 | version.dll | 0.0219 | libgimpbase-2.0-0.dll | 0.0335 |
| 12 | winmm.dll | 0.0218 | libgobject-2.0-0.dll | 0.0288 |
| 13 | msvcrt.dll | 0.0206 | libgimpui-2.0-0.dll | 0.0286 |
| 14 | comdlg32.dll | 0.0190 | libgtk-win32-2.0-0.dll | 0.0274 |
| 15 | shlwapi.dll | 0.0161 | oleaut32.dll | 0.0270 |
| 16 | mfc42.dll | 0.0134 | libgimpwidgets-2.0-0.dll | 0.0270 |
| 17 | msimg32.dll | 0.0121 | gdi32.dll | 0.0219 |
| 18 | winhttp.dll | 0.0101 | shlwapi.dll | 0.0187 |
| 19 | winspool.drv | 0.0084 | comctl32.dll | 0.0187 |
| 20 | gdiplus.dll | 0.0082 | ntdll.dll | 0.0181 |

**Table A2   The normalized frequency of top 20 API functions imported by malware and benign files.**

| No. | Malware | | Benign | |
|---|---|---|---|---|
| | API functions | Normalized frequency | API functions | Normalized frequency |
| 1 | getprocaddress | 0.0693 | corexemain | 0.0754 |
| 2 | corexemain | 0.0610 | getcurrentthreadid | 0.0548 |
| 3 | exitprocess | 0.0602 | getsystemtimeasfiletime | 0.0536 |
| 4 | loadlibrarya | 0.0565 | queryperformancecounter | 0.0532 |
| 5 | getlasterror | 0.0541 | getcurrentprocessid | 0.0532 |
| 6 | getcurrentprocess | 0.0530 | getcurrentprocess | 0.0514 |
| 7 | writefile | 0.0515 | exit | 0.0512 |
| 8 | sleep | 0.0512 | getlasterror | 0.0509 |
| 9 | multibytetowidechar | 0.0507 | sleep | 0.0497 |
| 10 | widechartomultibyte | 0.0507 | unhandledexceptionfilter | 0.0492 |
| 11 | getmodulehandlea | 0.0487 | terminateprocess | 0.0487 |
| 12 | getcurrentthreadid | 0.0464 | setunhandledexceptionfilter | 0.0486 |
| 13 | closehandle | 0.0458 | gettickcount | 0.0476 |
| 14 | unhandledexceptionfilter | 0.0442 | cexit | 0.0459 |
| 15 | gettickcount | 0.0440 | initterm | 0.0459 |
| 16 | leavecriticalsection | 0.0430 | setusermatherr | 0.0457 |
| 17 | entercriticalsection | 0.0429 | setapptype | 0.0457 |
| 18 | deletecriticalsection | 0.0428 | memcpy | 0.0440 |
| 19 | terminateprocess | 0.0420 | amsgexit | 0.0432 |
| 20 | getstdhandle | 0.0419 | getprocaddress | 0.0421 |

### Funding
The work was supported by the Korea Institute of Science and Technology under the KIST School Partnership Project for its alumni. The funders had no role in study design, data collection and analysis, decision to publish, or preparation of the manuscript.

### Grant Disclosures
The following grant information was disclosed by the authors:
Korea Institute of Science and Technology under the KIST School Partnership Project for its alumni.

### Competing Interests
The authors declare there are no competing interests.

### Author Contributions
- Muhammad Irfan Yousuf conceived and designed the experiments, performed the experiments, analyzed the data, performed the computation work, authored or reviewed drafts of the article, and approved the final draft.

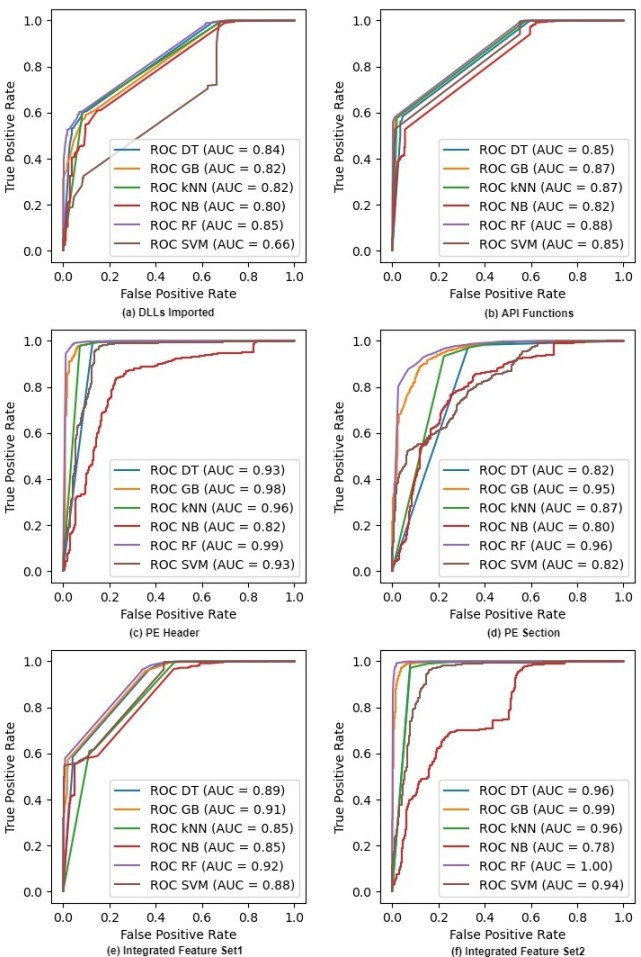

**Figure A1** ROC curves for different classifiers on selected feature sets after applying Information Gain.

- Izza Anwer conceived and designed the experiments, performed the experiments, prepared figures and/or tables, and approved the final draft.
- Ayesha Riasat conceived and designed the experiments, analyzed the data, performed the computation work, authored or reviewed drafts of the article, and approved the final draft.
- Khawaja Tahir Zia conceived and designed the experiments, performed the experiments, performed the computation work, prepared figures and/or tables, and approved the final draft.
- Suhyun Kim conceived and designed the experiments, analyzed the data, authored or reviewed drafts of the article, and approved the final draft.

## Data Availability

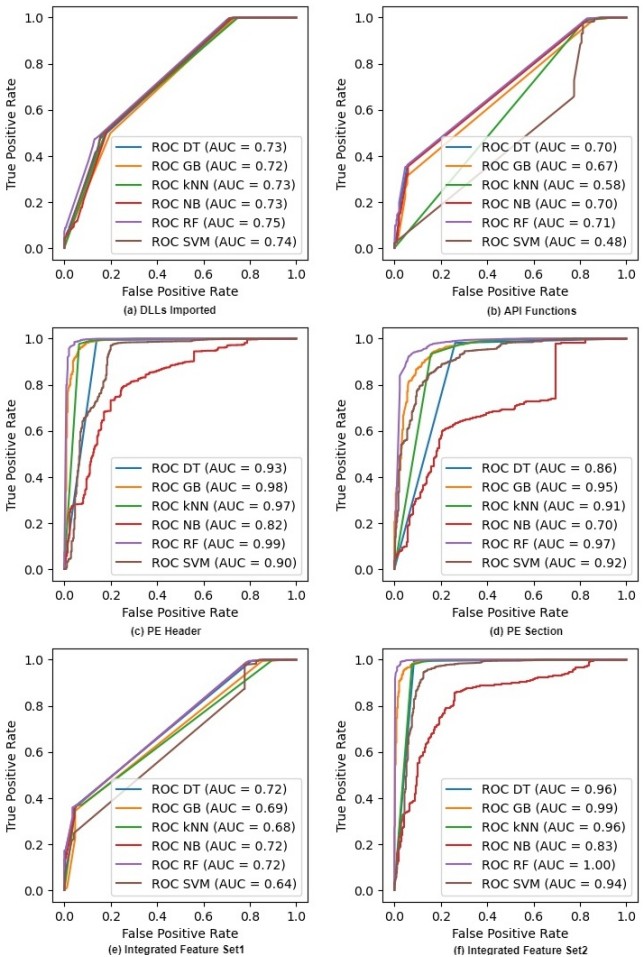

**Figure A2** ROC curves for different classifiers on selected feature sets after applying principal component analysis.

The data is available at FigShare: Yousuf, Irfan (2023): Windows Malware Detection Dataset. figshare. Dataset. https://doi.org/10.6084/m9.figshare.21608262.v1

## Supplemental Information

Supplemental information for this article can be found online at http://dx.doi.org/10.7717/peerj-cs.1319#supplemental-information.

# REFERENCES

**Alhaidari F, Shaib N, Alsafi M, Alharbi H, Alawami M, Aljindan R, Rahman A, Zagrouba R. 2022.** ZeVigilante: detecting zero-day malware using machine learning and sandboxing analysis techniques. *Computational Intelligence and Neuroscience* 1615528.

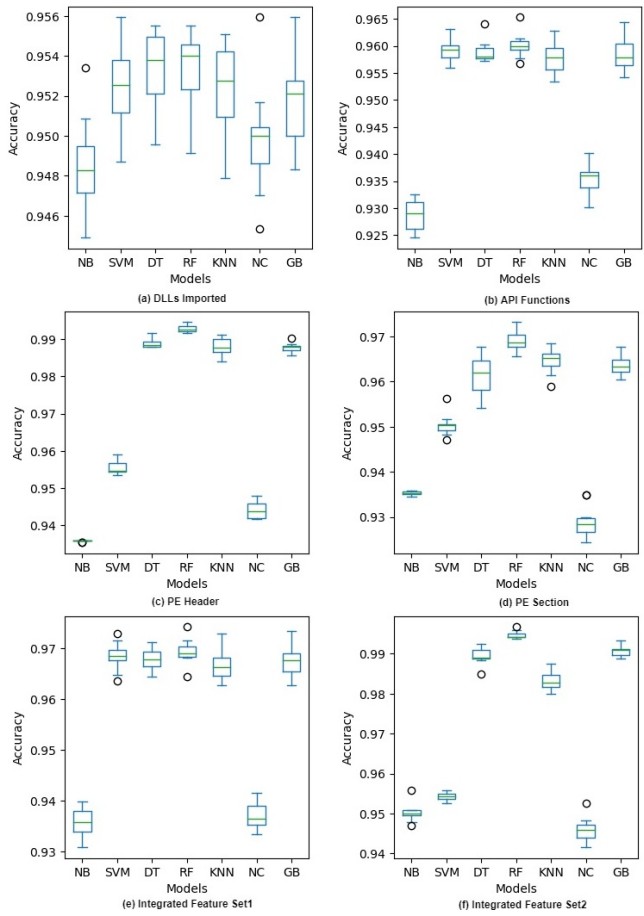

**Figure A3** Box plots for different classifiers on selected feature sets after applying Information Gain.

**Amer E, Zelinka I. 2020.** A dynamic Windows malware detection and prediction method based on contextual understanding of API call sequence. *Computers and Security* **92**:101760 DOI 10.1016/j.cose.2020.101760.

**Azmee AA, Choudhury PP, Alam M, Dutta O, Hossain MI. 2020.** Performance analysis of machine learning classifiers for detecting PE malware. *International Journal of Advanced Computer Science and Applications* **11(1)**.

**Cannarile A, Dentamaro V, Galantucci S, Iannacone A, Impedovo D, Pirlo G. 2022.** Comparing deep learning and shallow learning techniques for API calls malware prediction: a study. *Applied Sciences* **12(3)**:1645.

**Catak FO, Yazi A, Elezaj O. 2020.** Deep learning based Sequential model for malware analysis using Windows exe API calls. *PeerJ Computer Science* **6**:e285 DOI 10.7717/peerj-cs.285.

**Cepeda C, Chia Tien DL, Ordóñez P. 2016.** Feature selection and improving classification performance for malware detection. In: *2016 IEEE international conferences on big data and cloud computing (BDCloud), social computing and networking*

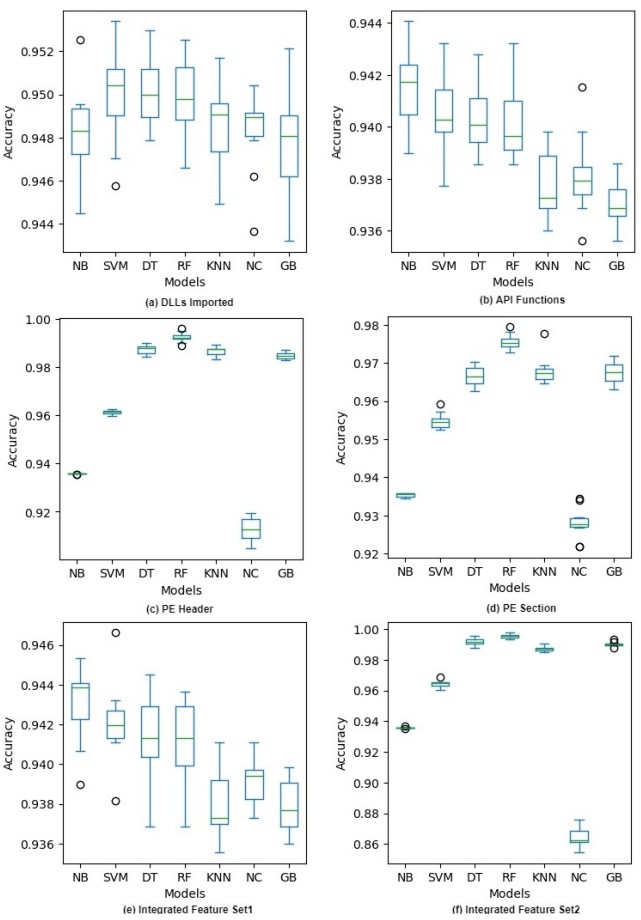

**Figure A4** Box plots for different classifiers on selected feature sets after applying principal component analysis.

(SocialCom), sustainable computing and communications (SustainCom) (BDCloud-SocialCom-SustainCom). Piscataway: IEEE, 560–566.

**Chowdhury M, Rahman A, Islam R. 2017.** Protecting data from malware threats using machine learning technique. In: *2017 12th IEEE conference on industrial electronics and applications (ICIEA)*. Piscataway: IEEE, 1691–1694.

**Damaševičius R, Venčkauskas A, Toldinas J, Grigaliūnas A. 2021.** Ensemble-based classification using neural networks and machine learning models for windows PE malware detection. *Electronics* **10(4)**:485.

**Euh S, Lee H, Kim D, Hwang D. 2020.** Comparative analysis of low-dimensional features and tree-based ensembles for malware detection systems. *IEEE Access* **8**:76796–76808 DOI 10.1109/ACCESS.2020.2986014.

**Fuyong Z, Tiezhu Z. 2017.** Malware detection and classification based on N-grams attribute similarity. In: *2017 IEEE international conference on computational science and engineering (CSE) and IEEE international conference on embedded and ubiquitous computing (EUC), vol. 1*. Piscataway: IEEE, 793–796.

**Gibert D, Mateu C, Planes J. 2020.** The rise of machine learning for detection and classification of malware: research developments, trends and challenges. *Journal of Network and Computer Applications* **153**:102526.

**Guo Y. 2023.** A review of machine learning-based zero-day attack detection: challenges and future directions. *Computer Communications* **198(C)**:175–185.

**Gupta D, Rani R. 2018.** Big data framework for zero-day malware detection. *Cybernetics and Systems* **49(2)**:103–121 DOI 10.1080/01969722.2018.1429835.

**Kim S, Yeom S, Oh H, Shin D, Shin D. 2021.** Automatic malicious code classification system through static analysis using machine learning. *Symmetry* **13(1)**:35.

**Kumar A, Kuppusamy K, Aghila G. 2019.** A learning model to detect maliciousness of portable executable using integrated feature set. *Journal of King Saud University—Computer and Information Sciences* **31(2)**:252–265 DOI 10.1016/j.jksuci.2017.01.003.

**Kumar S, Singh CBB. 2018.** A zero-day resistant malware detection method for securing cloud using SVM and sandboxing techniques. In: *2018 second international conference on inventive communication and computational technologies (ICICCT)*. Piscataway: IEEE, 1397–1402.

**Naval S, Laxmi V, Rajarajan M, Gaur MS, Conti M. 2015.** Employing program semantics for malware detection. *IEEE Transactions on Information Forensics and Security* **10(12)**:2591–2604 DOI 10.1109/TIFS.2015.2469253.

**O'Kane P, Sezer S, McLaughlin K. 2011.** Obfuscation: the hidden malware. *IEEE Security & Privacy* **9(5)**:41–47.

**Patidar P, Khandelwal H. 2019.** Zero-day attack detection using machine learning techniques. *International Journal of Research and Analytical Reviews* **6(1)**:1364–1367.

**Pedregosa F, Varoquaux G, Gramfort A, Michel V, Thirion B, Grisel O, Blondel M, Prettenhofer P, Weiss R, Dubourg V, Vanderplas J, Passos A, Cournapeau D, Brucher M, Perrot M, Duchesnay E. 2011.** Scikit-learn: machine learning in Python. *Journal of Machine Learning Research* **12**:2825–2830.

**Pham H-D, Le TD, Vu TN. 2018.** Static PE malware detection using gradient boosting decision trees algorithm. In: *International conference on future data and security engineering*. Cham: Springer, 228–236.

**Raff E, Barker J, Sylvester J, Brandon R, Catanzaro B, Nicholas CK. 2018.** Malware detection by eating a whole EXE. ArXiv preprint. arXiv:1710.09435.

**Sharma S, Rama Krishna C, Sahay SK. 2019.** Detection of advanced malware by machine learning techniques. In: *Soft Computing: Theories and Applications: Proceedings of SoCTA 2017*. 333–342.

**Singh J, Singh J. 2020.** Detection of malicious software by analyzing the behavioral artifacts using machine learning algorithms. *Information and Software Technology* **121**:106273 DOI 10.1016/j.infsof.2020.106273.

**Tang M, Qian Q. 2019.** Dynamic API call sequence visualisation for malware classification. *IET Information Security* **13(4)**:367–377 DOI 10.1049/iet-ifs.2018.5268.

**Venkatraman S, Alazab M. 2018.** Use of data visualisation for zero-day malware detection. *Security and Communication Networks* **2018**.

**Wojnowicz M, Chisholm G, Wolff M, Zhao X. 2016.** Wavelet decomposition of software entropy reveals symptoms of malicious code. *Journal of Innovation in Digital Ecosystems* **3(2)**:130–140 DOI 10.1016/j.jides.2016.10.009.

**Zhang S-H, Kuo C-C, Yang C-S. 2019.** Static PE malware type classification using machine learning techniques. In: *2019 international conference on intelligent computing and its emerging applications (ICEA)*. 81–86.

**Zhang Y, Chang X, Lin Y, Mišić J, Mišić VB. 2020.** Exploring function call graph vectorization and file statistical features in malicious PE file classification. *IEEE Access* **8**:44652–44660 DOI 10.1109/ACCESS.2020.2978335.

**Zhang Y, Liu Z, Jiang Y. 2022.** The classification and detection of malware using soft relevance evaluation. *IEEE Transactions on Reliability* **71(1)**:309–320 DOI 10.1109/TR.2020.3020954.