# Peer review of "Windows malware detection based on static analysis with multiple features"

_PeerJ Computer Science, doi:10.7717/peerj-cs.1319_

## Round 0.1 · original submission · Major Revisions

Authors are advised to address the comments carefully recommended by the reviewers and resubmit the paper.

Reviewer 1 ·

Basic reporting

Basic Reporting
The aim of the article is to “developing new methods to detect malware”. The authors started the introduction by providing some classification of malware. At this point (line#36-39) it is recommended to provide a clear introduction to the reader what is your research scope and why it is considered as trend research. Avoid writ “other classes” as stated in (line#37) or “other related areas (line#39), it is recommended to specify the most related areas instead of generalized the research.
The citations must be mentioned at some sections, example (line#40-46) doesn’t mentioned any citations. At (line#47), what does it means “de facto”?. At (line#49) double check the reference and ensure it meets with the sentence that is written here. At (line#55), mentioned that the aim is to “propose a malware detection system”, while at the first paragraph mention “developing new method to detect malware”. I think the authors has to be more specific weather the aims is to propose an comprehensive system to detect the malware” or “to propose a method/sub-system to be used within a system that is already in use now days”.
The main objectives as listed (line#58-63), required to be fit with the final expected outcome from this research, while the points are the steps to perform the research.
The structure of the research is good to be illustrated as shown at (line#64-69), but it is recommended to avoid writing the quotation mark at the section name.

Related Work
It is better to name this part as “Literature Reviews”. The authors illustrate some relevant studies that are conducted on the scope of detecting malware in windows environment, but what is the research gap that was not yet covered by previous research? Also, what is the overall conclusion from the part (Related Work)? In general, the literature review must be improved.

Conclusion
Improve this part by including status of achiving the research objectives, the main research gap, and the future researches.

Experimental design

Method
Recommended to rename this part to be “Research Method”. At (line#123-124), required a references that support the approach.

Validity of the findings

No comment

Additional comments

The topic is interesting to be published with the results. It is required major english proofreading and more supporting updated literatures to enhanced the contents and the used approach.

·

Basic reporting

The manuscript is well organized and structured. Good to read with no flaws in English or the way the research carried-out is presented.

Abstract is self-contained, perfect.

In the Introduction section, suggest to include (before line # 64) 'the research contributions made by the authors.' Must revise.

The Section, Related Work needs revision as follows:

1) Related work referred and cited should be recent, preferably less than 3 years. For e.g., Reference # 6, 12, 20, and 22 needs to be refreshed. Reference #22 is too old and it occurs to me as obsolete.

2) Related work should contribute rather than summarizing or listing who did what. For e.g., line #74 and 75 states, "The authors [22] classified malware by extracting least correlated features from portable executables. Likewise it goes the same for the entire section. The last two sentences of this Related Work section states (line# 119 - 121), "The study [27] focuses on malware type detection or classification of malware family instead of binary classification. The work [19] applies gradient boosting decision trees to detect malware in windows environment." and it ends there.

Related work should be organized by methods, or approaches, or idea, or theory etc. Should compare, contrast, that includes evaluating their pros and cons, then synthesize related work and the authors should make their own observation and their findings. All these are missing and must be included.

Experimental design

Observed no major concerns. Looks good.

Validity of the findings

In the section 'comparison with previous work'. Table 9 shows that the proposed work accuracy is 99.5% using Random Forest classifier. The existing works accuracy ranges from 98.3% to 98.7%. The improvisation in terms of accuracy of the proposed work when compared with the closest existing method is 0.8%. What "impact" this small percentage of improvisation the proposed method makes as an outcome of malware detection system? Is it worth while? Please include this discussion in the same section.

Additional comments

There are no additional comments except for the observations made above in this report that needs to be considered by the authors as revision.

Otherwise, the manuscript is good. The above revision suggested if considered would assist to strengthen the quality of this manuscript that the authors could be proud of when it gets published.

·

Basic reporting

The paper titled "Windows malware detection based on static analysis with multiple features", is a novel writing as so far there are articles on windows malware detection but using multiple features set and creating novel data set for experiments has never seen before. The paper is clearly written in a good style and includes figures and tables wherever necessary.

Experimental design

The purpose of the paper has been very well stated in the abstract but needs clarification on the following:
Why this particular Portable Executable (PE) malware is chosen for this work?

The objectives mentioned in the paper are very appropriate and the discussion to prove that the objectives have been clearly attained is satisfactory.
In the discussion section, the research's strengths, limitations, and generality are adequately discussed compared to the other researcher's work discussed in the introduction and literature review sections. The authors have clearly acknowledged and identified the contributions of their research against previous researchers' work.

Validity of the findings

The authors adequately evaluated their work, and all claims are clearly articulated and supported by empirical experiments.

Additional comments

However, addressing the above comments would improve the quality of the paper. The overall work is good, novel and timely.

---

## Round 0.2 · accepted · Accept

Congratulations! on the acceptance of your work. Check your manuscript carefully for typos before publication.

Reviewer 1 ·

Basic reporting

The authors improved the article, and just a point that is to reduce the use of "we" and "our".

Experimental design

'no comment'

Validity of the findings

'no comment'

---

## Author Rebuttal · Round 0.2

# Reply to Reviewers Comments

Reviewer 1 (Anonymous)

Basic reporting

## Comment # 1:
The aim of the article is to "developing new methods to detect malware". The authors started the introduction by providing some classification of malware. At this point (line#36-39) it is recommended to provide a clear introduction to the reader what is your research scope and why it is considered as trend research.

## Reply # 1:
We re-wrote the first paragraph (line#36-39) and added a few sentences at the end of this paragraph to highlight our research scope.

Our work is aimed at developing a static malware detection system to detect Portable Executable (PE) malware using multiple features. We not only extract multiple features from PE malware but also combine these features to create integrated features in a bid to improve the accuracy of our malware detection system. Presumably, the multiple and integrated features used in this work have never been considered together in detecting malware.

## Comment # 2:
Avoid writ "other classes" as stated in (line#37) or "other related areas (line#39), it is recommended to specify the most related areas instead of generalized the research.

## Reply # 2:
We re-wrote the first paragraph and avoided generalizing the research areas.

## Comment # 3:
The citations must be mentioned at some sections, example (line#40-46) doesn't mentioned any citations.

## Reply # 3:
We added 2 citations to support our writing. References 8, 12 are added

**Comment # 4:**

At (line#47), what does it means "de facto"?.

**Reply # 4:**

We rephrased line#47 as, "During the last decade, machine learning has solved many problems in different sectors including cyber security".

**Comment # 5:**

At (line#49) double check the reference and ensure it meets with the sentence that is written here.

**Reply # 5:**

Thank you for pointing out the mistake. It was supposed to be reference number [1] instead of [9]. We corrected it.

**Comment # 6:**

At (line#55), mentioned that the aim is to "propose a malware detection system", while at the first paragraph mention "developing new method to detect malware". I think the authors has to be more specific weather the aims is to propose an comprehensive system to detect the malware" or "to propose a method/sub-system to be used within a system that is already in use now days".

**Reply # 6:**

The aim of this work is to propose a malware detection system as mentioned in the abstract and every where else in the paper. We replaced the word 'method' with 'system' in the first paragraph to avoid any confusion.

**Comment # 7:**

The main objectives as listed (line#58-63), required to be fit with the final expected outcome from this research, while the points are the steps to perform the research.

**Reply # 7:**

We changed line#62-68 to highlight our research contribution.

**Comment # 8:**

The structure of the research is good to be illustrated as shown at (line#64-69), but it is recommended to avoid writing the quotation mark at the section name.

**Reply # 8:**

We removed the quotation marks.

**Comment # 9:**

Related Work

It is better to name this part as "Literature Reviews". The authors illustrate some relevant studies that are conducted on the scope of detecting malware in windows environment, but what is the research gap that was not yet covered by previous research? Also, what is the overall conclusion from the part (Related Work)? In general, the literature review must be improved.

**Reply # 9:**

We renamed it to Literature Review and added a paragraph to conclude this section and highlight the research gap.

In conclusion, there is a vast amount of research on malware detection using machine learning and deep learning. Upon reviewing prior studies, a few key points can be noted. Firstly, most prior works only utilize one or two raw features in their malware detection efforts. Only a limited number of studies have combined raw features to create new ones. Secondly, feature selection is usually done through either Information Gain or Principal Component Analysis, with few studies employing both methods. Thirdly, ensemble learning is not widely used in these studies. Our study, however, extracts four raw features and creates two integrated features. We also apply both Information Gain and Principal Component Analysis for feature selection, use seven different classifiers for malware classification, and incorporate three ensemble learning techniques to increase classification accuracy.

**Comment # 10:**

Conclusion

Improve this part by including status of achieving the research objectives, the main research gap, and the future researches.

**Reply # 10:**

We added the following two paragraphs to improve it.

As a tangible outcome, a preprocessed dataset having 27,920 malware samples is created and available on request along with raw and integrated feature sets for comparing future work with the proposed work. We tested the proposed malware detection system extensively and performed multiple experiments on raw and integrated features to check its performance. By applying two feature selection methods, seven machine learning classifiers and three ensemble learning techniques

on multiple features, we tried to bridge the gap in the previous works on malware detection.

Our experiments show that PE Header forms the best feature set and gives the maximum accuracy and minimum error rate when integrated with PE Section. However, real-world scenario can be different from the experimental environment, hence, we cannot recommend using PE Header alone to detect malware. But, we do assert that this could be a starting point to further explore PE Header and its fields to develop a feature set for detecting zero-day attacks accurately and quickly. In future studies, we can add more file formats such as image, pdf, audio, and video etc. We can also work on adding mobile environments such as iOS and android.

## Comment # 11:
Experimental design

Method
Recommended to rename this part to be "Research Method". At (line#123-124), required a references that support the approach.

## Reply # 11:
The part is renamed to Research Method. We believe that the approach is very general as almost all the studies first collect malware samples and then detect malware and does not require references.

Validity of the findings

No comment

## Comment # 12:
Additional comments

The topic is interesting to be published with the results. It is required major english proofreading and more supporting updated literatures to enhanced the contents and the used approach.

## Reply # 12:
We worked hard on improving the writing including the English language mistakes. The literature review section has also been improved by adding new references.

## Comment # 1:
Basic reporting

The manuscript is well organized and structured. Good to read with no flaws in English or the way the research carried-out is presented.

Abstract is self-contained, perfect.

In the Introduction section, suggest to include (before line # 64) 'the research contributions made by the authors.' Must revise.

## Reply # 1:
We added our research contribution (line#62-68)

The research contribution made by the authors is listed below.
1. Collection of latest samples to create a new dataset of PE malware and benign files.
2. We extract four feature sets including the list of imported DLLs and API functions called by these samples, values of 52 attributes from PE Header and 100 attributes of PE Section.
3. We merge extracted features for creating new integrated features of PE samples.
4. Comprehensive analysis and evaluation of different machine learning classifiers, ensemble learning and feature selection techniques to maximize the malware detection rate.

## Comment # 2:
The Section, Related Work needs revision as follows:

1) Related work referred and cited should be recent, preferably less than 3 years. For e.g., Reference # 6, 12, 20, and 22 needs to be refreshed. Reference #22 is too old and it occurs to me as obsolete.

2) Related work should contribute rather than summarizing or listing who did what. For e.g., line #74 and 75 states, "The authors [22] classified malware by extracting least correlated features from portable executables. Likewise it goes the same for

the entire section. The last two sentences of this Related Work section states (line# 119 - 121), "The study [27] focuses on malware type detection or classification of malware family instead of binary classification. The work [19] applies gradient boosting decision trees to detect malware in windows environment." and it ends there.

Related work should be organized by methods, or approaches, or idea, or theory etc. Should compare, contrast, that includes evaluating their pros and cons, then synthesize related work and the authors should make their own observation and their findings. All these are missing and must be included.

**Reply # 2:**

We improved the related work section (now Literature Review) by adding more details of previous works and comparing with the proposed work when concluding this section. However, it should be noted that it is not a review paper or a survey paper, therefore, we did not go into the very details of related work. We also focused on the previous works which either detect PE malware (not android or other types for example) or apply machine learning / deep learning methods on malware samples. We removed references 6, 12, 20, 22 (numbers are as per previous version of this manuscript) and added 5 new references as discussed below.

Y. Zhang et al.[1] argued that most malware solutions only detect malware families that were included in the training data. They proposed to use a soft relevance value based on multiple trained models. They used features such as file sizes, function call names, DLLs, n-grams, etc. When the models are trained, we try to predict which malware family from the dataset they belong to. By using the trained models, the soft relevance value is applied to find if the malware belongs to one of the original malware families or not.

Singh & Singh [2] proposed a behavior-based malware detection technique. Firstly, printable strings are processed word by word using text mining techniques. Secondly, Shannon entropy is computed over the printable strings and API calls to consider the randomness of API and finally, all features are integrated to develop the malware classifiers using the machine learning algorithms.

Cannarile et al. [3] presented a benchmark to compare deep learning and shallow learning techniques for API calls malware detection. They considered Random Forest, CatBoost, XGBoost, and ExtraTrees as shallow learning methods whereas TabNet and NODE (Neural Oblivious Decision Ensembles) were used as deep learning methods. Based on experimental results, they concluded that shallow learning techniques tend to perform better and converge faster (less training time) to a suitable solution.

Euh et al. [4] propose low-dimensional but effective features for a malware detection system and analyze them with tree-based ensemble models. They extract the five types of malware features represented from binary or disassembly files. The experimental work shows that the tree-based ensemble model is effective and efficient for malware classification concerning training time and generalization performance.

Amer et al. [5] introduce the use of word embedding to understand the contextual relationship that exists between API functions in malware call sequence. Their experimental results prove that there is a significant distinction between malware and goodware call sequences. Next, they introduce a new method to detect and predict malware based on the Markov chain.

[1] Y. Zhang, Z. Liu, and Y. Jiang, "The classification and detection of malware using soft relevance evaluation," IEEE Transactions on Reliability, pp. 1–12, 2020.
[2] Singh, J.; Singh, J. Detection of malicious software by analyzing the behavioral artifacts using machine learning algorithms. Inf. Softw. Technol. 2020, 121, 106273
[3] Cannarile, A.; Dentamaro, V.; Galantucci, S.; Iannacone, A.; Impedovo, D.; Pirlo, G. Comparing Deep Learning and Shallow Learning Techniques for API Calls Malware Prediction: A Study. Appl. Sci. 2022, 12, 1645
[4] Euh, S.; Lee, H.; Kim, D.; Hwang, D. Comparative Analysis of Low-Dimensional Features and Tree-Based Ensembles for Malware Detection Systems. IEEE Access 2020, 8, 76796–76808
[5] Amer, E.; Zelinka, I. A dynamic Windows malware detection and prediction method based on contextual understanding of API call sequence. Comput. Secur. 2020, 92, 101760

Experimental design

Observed no major concerns. Looks good.

**Comment # 3:**
Validity of the findings

In the section 'comparison with previous work'. Table 9 shows that the proposed work accuracy is 99.5% using Random Forest classifier. The existing works accuracy ranges from 98.3% to 98.7%. The improvisation in terms of accuracy of the proposed work when compared with the closest existing method is 0.8%. What "impact" this small percentage of improvisation the proposed method makes as an outcome of malware detection system? Is it worth while? Please include this discussion in the same section.

**Reply # 3:**
We improved Table 9 as follows and added / rephrased lines # 343-359

| Work | Accuracy | Error Rate | Classifier | Feature set |
|------|----------|-----------|-----------|-------------|
| Proposed Work | 99.5% | 0.47% | Random Forest | Integrated feature set |
| Kumar et al.[15] | 98.3% | 1.47% | Random Forest | Integrated feature set |
| Azmee et al. [3] | 98.6% | 1.41% | XGBoost | 77 features of PE header |
| Damaševičius et al.[8] | 98.7% | 1.32% | ExtraTrees | 68 features of PE header |
| Kim et al.[14] | 98.7% | 1.31% | AdaBoost | PE Structure |

The table shows that the proposed system produces a very small error. In other words, the probability of misclassification in the proposed system is much lower than the previous systems. We agree that in terms of accuracy the system improvement is marginal, however, when combined with other metrics, the proposed system gives better results especially in terms of very small error rate.

Additional comments

There are no additional comments except for the observations made above in this report that needs to be considered by the authors as revision.

Otherwise, the manuscript is good. The above revision suggested if considered would assist to strengthen the quality of this manuscript that the authors could be proud of when it gets published.

Reviewer 3 (POOJA KHERWA)

Basic reporting

The paper titled "Windows malware detection based on static analysis with multiple features", is a novel writing as so far there are articles on windows malware detection but using multiple features set and creating novel data set for experiments has never seen before. The paper is clearly written in a good style and includes figures and tables wherever necessary.

## Comment # 1:
Experimental design

The purpose of the paper has been very well stated in the abstract but needs clarification on the following:

Why this particular Portable Executable (PE) malware is chosen for this work?

The objectives mentioned in the paper are very appropriate and the discussion to prove that the objectives have been clearly attained is satisfactory. In the discussion section, the research's strengths, limitations, and generality are adequately discussed compared to the other researcher's work discussed in the introduction and literature review sections. The authors have clearly acknowledged and identified the contributions of their research against previous researchers' work.

## Reply # 1:
We added line # 155 - 158
The motivation for using PE files was arrived at by monitoring the submissions received over different malware databases. For example, more than 25% malware samples in malwarebazaar database are PE malware and make it the common file type for spreading malware. Similarly, 47.8% files submitted to Virustotal for analysis are PE files [15].

Validity of the findings

The authors adequately evaluated their work, and all claims are clearly articulated and supported by empirical experiments.

Additional comments

However, addressing the above comments would improve the quality of the paper. The overall work is good, novel and timely.